# The Utility of Nasal Challenges to Phenotype Asthma Patients

**DOI:** 10.3390/ijms23094838

**Published:** 2022-04-27

**Authors:** Guillermo Bentabol-Ramos, Rocio Saenz de Santa Maria-Garcia, Monica Vidal-Diaz, Ibon Eguiluz-Gracia, Almudena Testera-Montes

**Affiliations:** 1Pulmonology Unit, Hospital Regional Universitario de Malaga, 29010 Malaga, Spain; guille_br1993@hotmail.com (G.B.-R.); judasroson@gmail.com (M.V.-D.); 2Allergy Unit, Hospital Regional Universitario de Malaga, 29010 Malaga, Spain; rociossm.93@gmail.com (R.S.d.S.M.-G.); iboneguiluz@gmail.com (I.E.-G.); 3Allergy Research Group, Instituto de Investigacion Biomedica de Malaga (IBIMA) and RICORS “Enfermedades Inflamatorias”, 29010 Malaga, Spain

**Keywords:** asthma phenotypes, nasal challenge, biomarker, allergic asthma, NSAID-exacerbated respiratory disease

## Abstract

Asthma is a heterogeneous disease in terms of both phenotype and response to therapy. Therefore, there is a great need for clinically applicable tools allowing for improved patient classification, and selection for specific management approaches. Some interventions are highly helpful in selected patients (e.g., allergen immunotherapy or aspirin desensitization), but they are costly and/or difficult to implement. Currently available biomarkers measurable in peripheral blood or exhaled air display many limitations for asthma phenotyping and cannot identify properly the specific triggers of the disease (e.g., aeroallergens or NSAID). The united airway concept illustrates the relevant epidemiological and pathophysiological links between the upper and lower airways. This concept has been largely applied to patient management and treatment, but its diagnostic implications have been less often explored. Of note, a recent document by the European Academy of Allergy and Clinical Immunology proposes the use of nasal allergen challenge to confirm the diagnosis of allergic asthma. Similarly, the nasal challenge with lysine acetylsalicylate (L-ASA) can be used to identify aspirin-sensitive asthma patients. In this review, we will summarize the main features of allergic asthma and aspirin-exacerbated respiratory disease and will discuss the methodology of nasal allergen and L-ASA challenges with a focus on their capacity to phenotype the inflammatory disease affecting both the upper and lower airways.

## 1. Introduction

Asthma is a common inflammatory disease of the airways affecting 5–10% of the Western population [1]. Heterogeneity and differential response to therapy are two main characteristics defining the condition, especially in difficult-to-treat and severe patients. This complexity illustrates the need for asthma classification, not only regarding severity, but also based on the different pathogenic phenomena that underlie the inflammation and the bronchial remodelling (endophenotypes). The phenotyping of asthma patients is crucial for a tailored management of the condition, allowing for better health-related outcomes in individual patients.

Building on earlier classifications (extrinsic and intrinsic asthma), nowadays asthma is divided into phenotypes with or without T2 inflammation. T2 inflammation in asthma is defined by the infiltration of the bronchial mucosa by Th2 and IgE+B lymphocytes, group 2 innate lymphoid cells (ILC2), eosinophils, mast cells and basophils, together with the deleterious effects of type 2 mediators (IL-4, IL-5, IL-9, IL-13, and epithelial-derived cytokines like thymic stromal lymphopoietin) [2]. T2 asthma comprises two phenotypes based on the presence or absence of clinically-relevant sensitization to aeroallergens (namely allergic and eosinophilic non-allergic asthma). Within the T2 group, we observe different clinical manifestations and particularities defining subphenotypes as aspirin-exacerbated respiratory disease (AERD) (a subgroup of eosinophilic non-allergic asthma) or allergic bronchopulmonary aspergillosis, among others. Conversely, non-T2 asthma is a heterogeneous category defined by the absence of T2 inflammation [3].

To identify asthma phenotype, we need biomarkers which preferably are reliable and accessible for clinical practice. In asthma, many of them have been investigated, but only a few are currently used. At present, blood and sputum levels of eosinophils, serum levels of total IgE and fractional exhaled nitric oxide (FeNO) are the most used biomarkers because of their accessibility, correlation with T2 inflammation and prognostic value. As they reflect inflammation, their values vary depending on many factors, (e.g., corticosteroids decrease eosinophil levels). This variability can be an advantage, as some of them can be used in disease monitoring [4], or a disadvantage, as it can lead to false negatives results. On the other hand, skin prick test (SPT) and serum allergen-specific IgE are essential biomarkers in the diagnostic approach for allergic asthma. Nevertheless, they only denote sensitization which implies that often they are not sufficient to establish whether asthma symptoms in a specific patient are driven by the exposure to aeroallergens [5]. The capacity of each biomarker to identify the allergic triggers of asthma in the clinic is described in Figure 1.

In contrast, more specific and sensitive tests can be applied the clinical practice. One of them is a group of in vivo provocation tests (oral, nasal, or bronchial) which are the gold standard to identify the external specific triggers of the disease, for example, allergic asthma or AERD. Nevertheless, oral, or bronchial challenges are laborious procedures requiring appropriate facilities and trained personnel. Moreover, they can induce severe side effects, especially in patients with poor baseline conditions. On the other hand, nasal challenges are considerably shorter and safer. The use of nasal challenges for the phenotyping of asthma patients builds on the conceptual framework of the united airway concept. Figure 2 shows the utility and the information provided by the different asthma biomarkers available in the clinic. In the present review, we summarize allergic asthma and AERD phenotypical characteristics, the difficulties in their diagnosis, and the utility of nasal provocations to identify these conditions.

## 2. Allergic Asthma: A Diagnostic Challenge for Clinicians

Allergic asthma is the most common asthma phenotype, affecting more than 80% and 50% of children and adults with asthma, respectively [6,7,8]. Allergic asthma is defined by the presence of asthma and sensitization to aeroallergens, requiring that at least one of the sensitizations is clinically relevant [7,9]. Exposure to aeroallergens in allergic asthma patients triggers T2 inflammation involving cytokines like IL-5, IL-4, and IL-13. IL-4 and IL-13 are essential for isotype switching of B lymphocytes towards sIgE. In addition to these cytokines, other important inflammatory cells in allergic asthma are mast cells, CD4^+^ T cells, and ILC2 [10].

Clinically, allergic asthma usually begins in childhood and is often associated with other entities such as allergic rhinitis and/or atopic dermatitis [11,12]. It is more common in the male gender and patients with a family history of asthma or atopy [13]. Allergic asthma can produce intermittent or perennial symptoms, but patients with allergic asthma frequently develop seasonal exacerbations [14,15].

By definition, patients with allergic asthma must be sensitized to at least one aeroallergen and this sensitization must be clinically relevant. SPT or serum sIgE indicate atopy, but their mere positivity has no diagnostic value for allergy. The demonstration of allergy is more complex and the size of the papule in the SPT and/or sIgE levels in serum do not directly correlate with the relevance of the sensitization [16,17]. In patients with intermittent or mild persistent asthma, the clinical history (e.g., seasonality or variations in symptom severity) is often sufficient to establish the diagnosis of allergic asthma. However, in patients with moderate-to-severe persistent asthma (especially in those who are polysensitized), demonstrating the clinical relevance of atopic sensitization presents more difficulties. In these cases, in vivo tests such as nasal allergen challenge (NAC) and/or bronchial allergen challenge (BAC) may be required. These tests assess the nasal and bronchial response to the allergen, respectively, in a controlled manner [10]. However, the BAC is considered a research tool only, and its use in clinical practice is not recommended, beyond the diagnosis of occupational allergic asthma. This fact derives from the possibility of a late asthmatic response and the need for long-term discontinuation of inhaled corticosteroids for its performance. Although the European Academy of Allergy and Clinical Immunology has recently proposed a simplified methodology addressing these issues, several safety and cost-efficiency aspects prevent the BAC to be implemented in the clinical practice for asthma patients [18].

Importantly, the diagnosis of allergic asthma allows us to establish therapeutic strategies such as allergen avoidance or treatment with allergen immunotherapy (AIT), the latter being the only treatment that modifies the natural course of the disease. Interestingly, sublingual immunotherapy with house dust mites is indicated in patients with mild-to-moderate allergic asthma, even in those with partially controlled disease [19]. Moreover, omalizumab (an anti-IgE monoclonal antibody) is indicated in severe allergic asthma with sensitization to perennial allergens and decreased lung function [20]. Nevertheless, the success of these interventions is greatly dependent on the correct selection of candidate patients. Therefore, the discrimination between atopic non-allergic and allergic asthmatics is an unmet need in the clinic.

## 3. Aspirin-Exacerbated Respiratory Disease: A Diagnostic Challenge for Clinicians

AERD, also called non-steroidal anti-inflammatory drugs (NSAID)-exacerbated respiratory disease (NERD), is an acquired inflammatory disorder caused by a dysregulated response to NSAIDs. Firstly, it was described by Widal et al. [21] and it was followed by the so-called Samter triad (comprising chronic rhinosinusitis with nasal polyps (CRSwNP), asthma and NSAID sensitivity) [22]. It appears over the age of thirty-forty in patients with pre-existing asthma and CRSwNP, and it is slightly more prevalent in women. These individuals develop airway symptoms (nasal, sinusal and/or bronchial) 30–120 min after the intake of NSAIDs of any pharmacological group. A meta-analysis by White AA et al. described a prevalence in the asthmatic population of around 7% and 15% in patients with severe asthma [23].

AERD molecular pathogenesis is based on the dysregulation of arachidonic acid metabolism and its transition to eicosanoids (prostaglandins and leukotrienes) in cells involved in T2 inflammation (especially eosinophils). Arachidonic acid is converted into prostaglandins mostly through cycloxygenase-1 (COX-1) and into leukotrienes (especially LTE4) through reactions catalysed by the enzyme lipoxygenase (LOX). Thus, inhibition of intracellular COX-1 by an NSAID (strong COX-1 blocker like aspirin, pirazolone or aril-acetic and aril-propionic acids) decreases prostaglandins synthesis and increases levels of arachidonic acid metabolised by the LOX pathway [24]. In healthy subjects or individuals with mild-to-severe T2 airway inflammation, eosinophils can respond to the increased metabolic demands caused by NSAID intake and catabolize quickly the excess LTE4, thus preventing symptom onset. Conversely, the eosinophils from CRSwNP and asthma patients with very severe T2 inflammation are exhausted due to a high baseline activation status, and they have lost the capacity to respond to additional metabolic demands. Therefore, increased LTE4 generates a pro-inflammatory environment associated with bronchoconstriction, mucus hypersecretion and hypervascularisation. AERD patients present prominent T2 inflammation leading to increased eosinophils in the bronchial and nasosinusal mucosa, overexpression of platelets and increased mast cell degranulation [25]. AERD is a relevant risk factor associated with severe airway disease and treatment failure in CRSwNP and asthma. Indeed, aspirin sensitivity is a strong predictor of disease relapse after nasosinusal endoscopic surgery in CRSwNP patients [26].

Asthma presence may begin up to 2 years after the onset of nasosinusal symptoms in AERD. Patients with AERD often have more severe asthma than in other phenotypes, with worse response to standard therapies, numerous exacerbations, and lower pulmonary function with robust eosinophilic-T2 inflammation. Due to this inflammatory pattern, it is not exceptional that AERD patients are also atopic, but classically their sensitizations to aeroallergens are not clinically relevant.

Despite several attempts to find a valid biomarker for AERD, confirmation diagnosis can currently only be made by a specific NSAID challenge. Some studied biomarkers are serum periostin, dipeptidyl-peptidase 10 (DPP10) or urine LT4E. However, none of the above are sufficiently validated to be feasible in routine clinical practice. On the other hand, oral or bronchial challenge with aspirin is exempt from risk, especially in patients with a severe decrease of FEV1. Nevertheless, the identification of AERD patients has relevant implications for management. On the one hand, it permits establishing avoidance measures for strong COX-1 blockers, together with the need for oral provocations with selective COX-2 blockers (which are usually tolerated). On the other hand, it also allows identifying patients who will benefit from specific therapies like aspirin desensitization.

## 4. United Airway and Its Implications

The nasal, sinusal and bronchial mucosae show a close relationship both during homeostasis and inflammation, and this connection is clear at both the epidemiological and pathophysiological levels [27,28,29]. The European Community Respiratory Health Survey demonstrated that the presence of nasal symptoms multiplies by 3–4 the risk of asthma [30]. Of note, 40% and 70% of chronic rhinitis and CRSwNP patients, respectively, suffer from asthma, whereas up to 80% of asthmatics report nasosinusal complaints [31]. In AR patients without asthma, the NAC induces a decrease in lung and nasal volumes, together with an eosinophilic infiltrate detectable in both the nasal and bronchoalveolar lavage [32]. Importantly, atopic patients with rhinitis and asthma show a concordance >80% in the results of the NAC and BAC [33,34]. Moreover, the nasal exploration of asthmatic patients who do not report nasal symptoms, often reveals inflammatory changes in the nostril or paranasal sinuses [35]. All this evidence was integrated more than 20 years ago, within the so-called “united airway” concept [27].

This concept has relevant diagnostic and therapeutic implications [35]. Of note, it is possible to phenotype asthma patients based on the results of nasal tests [18]. Of note, as compared to bronchial specimens, nasal samples (e.g., lavage or biopsy) are easier to collect [19,28,36]. Moreover, nasal challenges are significantly shorter and safer than bronchial provocations [37]. Thus, nasal tests display many advantages for their clinical implementation [38]. Moreover, most interventions for airway diseases induce a clinical benefit in both the upper and lower airways [39]. Aspirin desensitization improves both CRSwNP and asthma outcomes [40], whereas allergen avoidance measures and AIT can control the symptoms and reduce the need for symptomatic medication in both allergic rhinitis and allergic asthma [39]. The later treatment also prevents asthma onset in allergic rhinitis patients and the appearance of new sensitizations to aeroallergens in atopic individuals [41]. On the other hand, the three biologicals indicated for severe CRSwNP (omalizumab, mepolizumab and dupilumab) have a pre-existing indication for asthma [42], whereas benralizumab and reslizumab display better performance in asthma patients with comorbid CRSwNP [10]. In this regard, there is a positive correlation between the degree of control of allergic rhinitis and allergic asthma [27], and the medical or surgical treatment of CRSwNP is connected to better control of severe asthma patients [40]. Finally, aspirin desensitization improves asthma and CRSwNP control in AERD patients and reduces the rate of relapse after nasosinusal endoscopic surgery [40].

## 5. Nasal Allergen Challenge: Methodology and Utility to Phenotype the United Airway Disease

The NAC is a diagnostic tool that reproduces in a controlled manner the response of the nasal mucosa to allergen exposure. It is a safe, reproducible, and cheap test [43]. The NAC has clinical indications to study patients in whom there is a discrepancy between the clinical history and the result of SPT and/or serum sIgE measurement, especially in polysensitized individuals in whom the initiation of AIT is considered; to confirm the diagnosis of local allergic rhinitis (symptoms triggered by the allergen in the absence of atopy), or of occupational allergic rhinitis [44]. In addition, NAC also has research applications such as studying the immunopathological mechanisms involved in allergic rhinitis or monitoring the effect of AIT in clinical trials [45,46]. However, NAC continues to be poorly implemented in clinical practice due to the absence of standardized protocols, among others. For this reason, the European Academy of Allergy and Clinical Immunology published a position paper in 2018 with the aim of harmonizing NAC for use in daily clinical practice [45].

NAC should be performed in the morning by trained medical personnel. It should be checked that the washout periods for drugs such as antihistamines or nasal corticosteroids, among others, have been carried out correctly. The patient must acclimatize for 15 min to the room conditions in which the NAC is going to be performed. After this, a baseline measurement is performed, followed by a control measurement to rule out nasal hyperreactivity and finally the measurements after the application of the allergen. Generally, a first measurement is carried out 15 min after allergen administration, with one or two additional assessments up to 60 min after allergen application. For diagnostic purposes, a single allergen dose (full concentration) is usually given. The concentration will rely on the allergen and extract used, as their biological power is usually not comparable. Although existing guidelines recommend only 1 allergen/session, our group described a NAC protocol with multiple allergens in one day, which allows for quick discrimination between local allergic and non-allergic patients [47]. It is recommended to apply the allergen bilaterally by means of a micropipette or spray vial with a 50 μL/puff nozzle. It is recommended to evaluate the response through symptom score (Lebel score, visual analogue scale [VAS], etc.) and the measurement of nasal patency (acoustic rhinometry [AcRh], active anterior rhinomanometry, etc.). European guidelines consider NAC positive if either a moderate change in both parameters (increase >23 mm^−3^ points for VAS/symptom score AND decrease >20–27% for nasal patency) or a clear change in at least one of them (increase >55/5 points for VAS/symptom score AND/OR decrease >40% for nasal patency) occur [45].

Following the united airways concept, in recent years NAC has been proposed as a tool to assess the inflammation that allergens produce throughout the airway (upper and lower). Braunsthal et al. demonstrated the presence of bronchial inflammation after NAC and nasal inflammation after BAC [32]. Pelikan later suggested that inflammatory mediators from the nasal mucosa might propagate to the lower airway after performing NAC [48]. Subsequently, other studies carried out by different groups have shown that NAC induces bronchial inflammation measurable through non-invasive methods (nasal lavage, condensed exhaled air, FeNO, etc.) without producing serious adverse reactions [49,50,51].

In summary, the study of inflammation induced by NAC in the lower respiratory tract is considered a promising tool in the diagnosis of allergic asthma, reducing the risk of serious adverse reactions that can be associated with BAC. Figure 3 shows a diagnostic algorithm for asthma, whereas Table 1 shows a comparison among NAC, BAC, and conjunctival allergen challenges.

## 6. Nasal Challenge with L-ASA: Methodology and Utility to Phenotype the United Airway Disease

The main indication for the nasal challenge with lysine-acetylsalicylate (NC-L-ASA) is the investigation of AERD in an asthma/CRSwNP patient who experienced airway symptoms (nasal and/or bronchial) after the intake of NSAIDs (especially strong COX-1 blockers) [52]. If the patient had ≥2 reactions with NSAIDs belonging to different pharmacological groups (e.g., aril-propionic, and aril-acetic acids), the diagnosis of NSAID intolerance is considered proven, with no need for additional tests [53]. However, in patients with a single airway reaction, further investigation is required [52]. The NC-L-ASA is especially indicated in those individuals where an oral or bronchial challenge cannot be performed (e.g., those with uncontrolled asthma or FEV1 < 70%) [53]. Nevertheless, it might represent the safest and most rapid alternative for AERD screening in the heterogeneous population of asthmatics seen in the clinic [40]. In this regard, NC-L-ASA displays an optimal specificity and PPV for AERD diagnosis, although the specificity and NPP are sub-optimal [53]. Therefore, negative results require confirmation by oral aspirin provocation [52].

Similar to NAC, the response to NC-L-ASA is measured by both subjective parameter (symptom score) and objective parameter (objective measurement of the nasal patency by AcRh or rhinomanometry) [52]. Additionally, forced spirometry is generally recommended before and after NC-L-ASA to assess lung function [53]. The washout period for the different anti-inflammatory drugs and the preliminary considerations (need for acclimation period for the patient) are the same as for NAC [52]. Regarding test performance, 0.1 mL of L-ASA is applied intra-nasally at the head of the lower turbinate with a syringe, pipette, or dropper [53]. Alternatively, ketorolac can be used for nasal provocation [52]. There is no consensus about the number of doses (1–4 administrations, with 18–20 mg as the maximum cumulative aspirin-equivalent dose) that should be given, although the administration of a single full dose (e.g., 14 mg) seems more acceptable than in oral or bronchial provocation [53]. Another aspect requiring further investigation is the optimal interval to monitor the response [52]. Besides the baseline assessment, it is mandatory to measure the response 30 min after the administration of the diluent [53]. This measurement is needed to control for nasal hyperreactivity [53]. Thereafter, most protocols monitor the response at 30 and 60 min after L-ASA administration [52]. Because airway reactions in AERD tend to occur later than IgE-mediated reactions in respiratory allergy, the measurement at 15 min is less often conducted than in NAC [53]. NC-L-ASA monitored by AcRh is considered positive when nasal volume decreases ≥25% as compared to post-diluent assessment at any measurement performed after L-ASA administration [52]. On the other hand, there is a lack of consensus on the optimal cut-off points for rhinomanometry yet an increase ≥100% of airflow resistance has been proposed [52]. Although the monitoring of nasal symptoms (e.g., by visual analogue scale) is encouraged by all published protocols, it is unclear how to integrate this parameter in the interpretation of the test [53]. NC-L-ASA is regarded as a very safe technique and complications are rare [52]. Nevertheless, bronchospasm is more frequent in NC-L-ASA than in NAC [53]. This fact probably derives from the greater baseline severity of asthma patients subjected to the former as compared to the later test [40]. Delayed reactions are also very uncommon and hard to interpret. Therefore, the general recommendation is to repeat the NC-L-ASA in case they are encountered [53].

AERD is a prototypical condition illustrating the united airway concept [53]. In most cases, patients suffer from severe eosinophilic inflammation affecting equally the nasal, sinusal and bronchial mucosae [52]. Commonly, the disease is as hard to control in the upper as compared to the lower airways [40]. Moreover, NSAID intake induces parallel reactions in the nasosinusal and bronchial compartments. Therefore, the positivity of the nasal provocation with L-ASA implies that both CRS and asthma are aspirin-sensitive in a specific patient [52]. Of note, this demonstration has key consequences for management and treatment [53]. The same rationale would apply for the bronchial provocation with L-ASA, but for safety and cost-efficiency considerations, the performance of NC-L-ASA offers much more advantages [52]. Figure 4 shows a diagnostic algorithm for asthma, whereas Table 2 shows a comparison among oral, bronchial, and nasal challenges with NSAIDs.

## 7. Conclusions

Asthma is a complex syndrome involving many phenotypes and subphenotypes. Nevertheless, the dependence and affection by external agents (always by unspecific stimuli like viral infections or pollution, and sometimes by specific triggers like allergens or NSAID) is a hallmark of the entity. This fact is illustrated by the recent COVID-19 pandemic when, despite a decreased accessibility to healthcare resources, most asthma patients reported an improved health status probably resulting from sheltering at home and the protection provided by facial masks. In any case, the identification of the environmental triggers of asthma has key implications for patients’ treatment and management.

This identification can hardly be done through in vitro biomarkers measurable in biological samples which are easily collected and processed. For example, the measurement of sIgE in serum has many limitations for the identification of allergic asthma patients and the selection of patients for AIT. On the other hand, in vivo provocations with allergens and NSAID represent the gold standard for the identification of specific disease triggers in asthma. Nevertheless, oral, or bronchial challenges are time-consuming and laborious techniques which are not exempt from risk, especially in severe asthma patients. Interestingly, nasal challenges display many advantages, as they can identify the specific triggers of asthma sensitively and are considerably shorter and safer than other in vivo provocations. Of note, the united airway concept does not only show epidemiological and therapeutic implications, but also has relevant consequences for patient diagnosis and phenotyping.

## Figures and Tables

**Figure 1 ijms-23-04838-f001:**
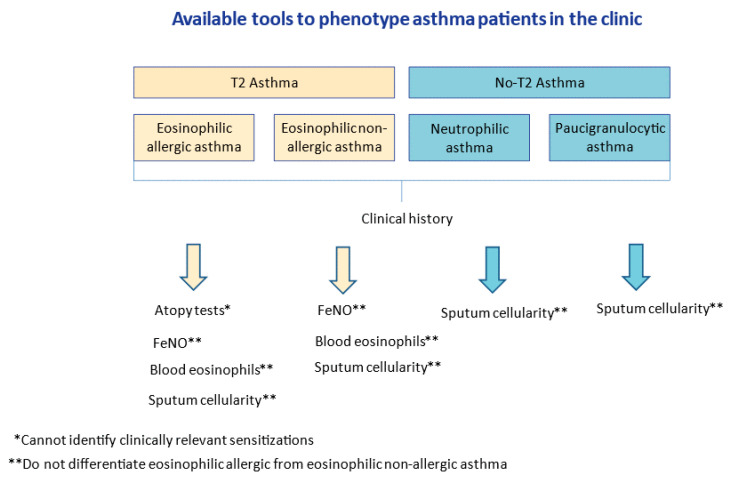
Description of asthma phenotypes, and utility of currently available biomarkers for the confirmation of allergic asthma in the clinic. FeNO: fractional exhaled nitric oxide; T2: type 2 inflammation.

**Figure 2 ijms-23-04838-f002:**
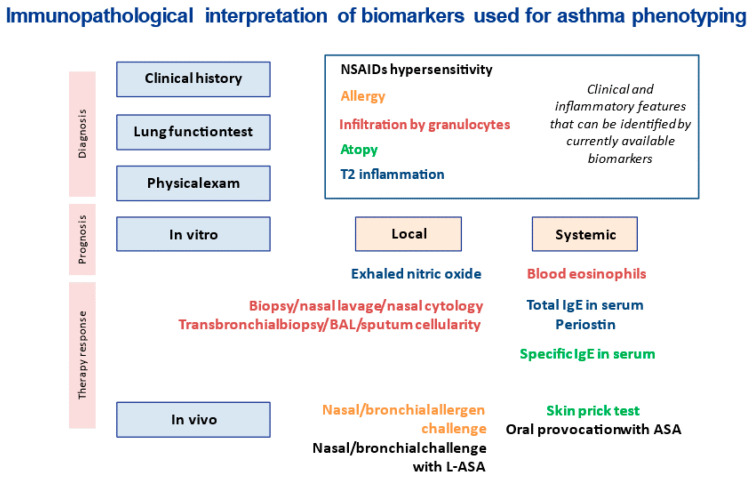
Immunopathological interpretation of biomarkers used for asthma phenotyping. ASA: acetylsalicylic acid; BAL: bronchoalveolar lavage; IgE: immunoglobulin E; L-ASA: lysine-acetylsalicylic acid; T2: type 2 inflammation. NSAID: non-steroidal anti-inflammatory drug.

**Figure 3 ijms-23-04838-f003:**
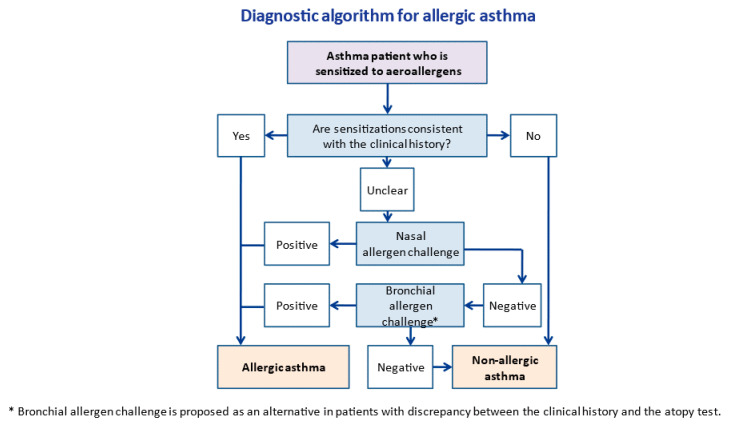
Diagnostic algorithm for allergic asthma.

**Figure 4 ijms-23-04838-f004:**
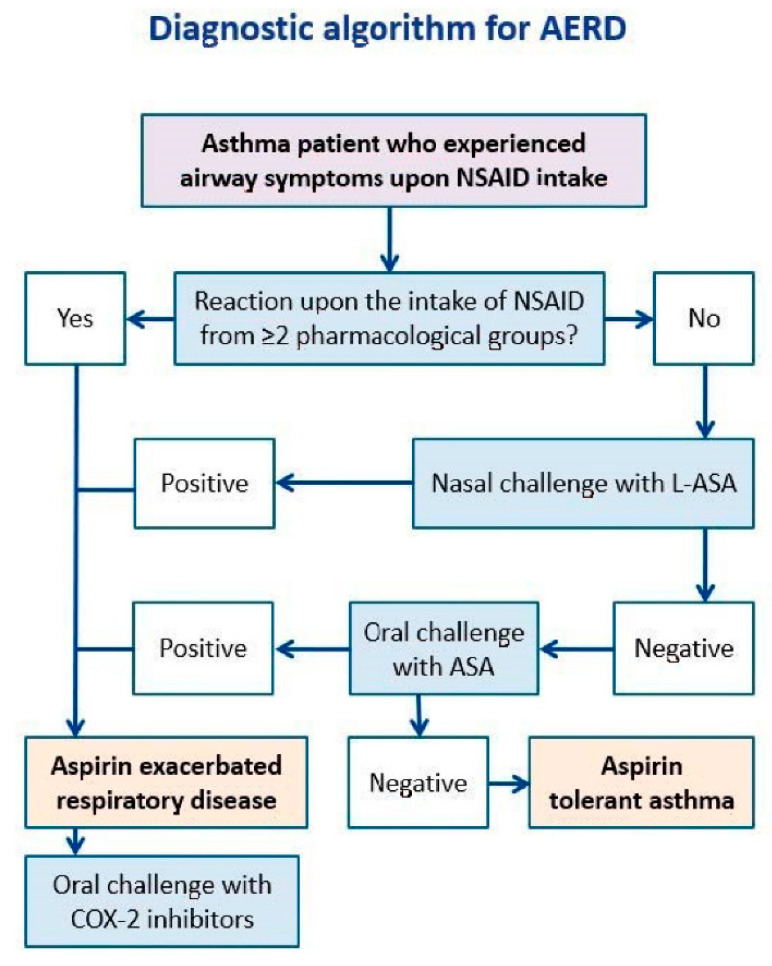
Diagnostic algorithm for aspirin-exacerbated respiratory disease. ASA: acetylsalicylic acid; L-ASA: lysine-acetylsalicylic acid; NSAID: non-steroidal anti-inflammatory drug.

**Table 1 ijms-23-04838-t001:** Differences between the nasal and bronchial allergen challenges. FEV1: forced expiratory volume in the 1st second; PNIF: peak nasal inspiratory flow.

	Nasal Allergen Challenge	Bronchial Allergen Challenge
**Indications**	- Diagnosis of local allergic rhinitis, occupational allergic rhinitis, allergic rhinitis in patients with a discrepancy between symptom pattern and atopy tests and monitoring of the efficacy of AIT	- Diagnosis of allergic asthma in patients with a discrepancy between symptom pattern and atopy tests, and occupational allergic asthma
**Absolute contra-indications**	- Acute inflammation of the nose or paranasal sinuses (<2 weeks) - Uncontrolled severe asthma or other pulmonary diseases - Systemic immunotherapy before NAC	- Uncontrolled or partially controlled asthma - FEV1 < 70% - Contraindication of inhaled β2 agonists, corticosteroids, or epinephrine - Acute respiratory infections
**Procedure**		
❖ **Allergen application**	Bilateral application (pump-metered aerosol spray, micropipette, or impregnated disk) Single-dose of undiluted allergen is recommended for the daily practice One allergen or multiple allergen provocation	Inhalation using a dosimeter at tidal breathing or through counted deep breaths It is recommended the inhalation of 5 increasing allergen concentrations One allergen per session
❖ **Monitoring**	Combination of subjective evaluation (symptom score) and objective evaluation (nasal patency)	FEV1 as measured by forced spirometry
**Limitations**	- Nasal hyper-reactivity - Poor standardization of some allergen extracts such as those of animal epithelia - Need for a relatively preserved nasal anatomy	- Bronchoconstriction induced by BAC - Transient increase in symptoms—Limited number of standardized allergen extracts for BAC
**Safety**	Late reactions are very rare After a positive NAC: observation period of 1 h at the hospital	Late reactions can occur After a positive BAC: observation period of 7 h at the hospital

**Table 2 ijms-23-04838-t002:** Comparison of the oral challenge with ASA and the bronchial and nasal challenge with L-ASA. ASA: acetylsalicylic acid; FEV1: forced expiratory volume in the 1st second; L-ASA: lysine-acetylsalicylic acid; NSAID: non-steroidal anti-inflammatory drug; PNIF: peak nasal inspiratory flow.

	Nasal Challenge with L-ASA	Bronchial Challenge with L-ASA	Oral Challenge with ASA
**Indications**	- Diagnosis of aspirin-exacerbated respiratory disease (especially when there is an involvement of the upper airways in the reactions experienced by the patient)	- Diagnosis of aspirin-exacerbated respiratory disease (especially when there is an involvement of the lower airways in the reactions experienced by the patient)	- Diagnosis of NSAID cross-intolerance (including aspirin-exacerbated respiratory disease)
**Contra-indications**	**Absolute contraindications:** - Acute inflammation of the nose or paranasal sinuses (<2 weeks) - Severe comorbidities and/or severe systemic diseases - Uncontrolled severe asthma or other pulmonary diseases - Pregnancy **Relative contraindications:** - Children under 5 years old - Temporary contraindications: - Acute allergic reactions in other organs - Recent vaccination (wait 1 week) - Acute viral or bacterial infection (wait 4 weeks) - Surgery of the nose or paranasal sinuses (wait 6–8 weeks) - Recent use of alcohol or tobacco for 24–48 h before NAC	**Absolute contraindications:** - Uncontrolled or partially controlled asthma - FEV1 < 70% - Pregnancy - Patients who have an absolute contraindication for the administration of inhaled β2 agonists, corticosteroids, or epinephrine - Inability to cooperate - Acute respiratory infections **Relative contraindications:** - Patients who cannot discontinue temporarily the intake of non-selective β blockers - Patients who cannot perform reproducible spirometry manoeuvres - Unstable cardiac, respiratory, immunologic, oncologic, or other important systemic diseases	**Absolute contraindications:** - Uncontrolled or partially controlled asthma - FEV1 < 70% - Pregnancy - Patients who have an absolute contraindication for the administration of inhaled β2 agonists, corticosteroids, or epinephrine - Inability to cooperate - Acute respiratory infections **Relative contraindications:** - Patients who cannot discontinue temporarily the intake of non-selective β blockers - Patients who cannot perform reproducible spirometry manoeuvres - Unstable cardiac, respiratory, immunologic, oncologic, or other important systemic diseases
**Procedure**			
❖ **Control test**	Bilateral application of isotonic pH neutral non-irritant solution before applying L-ASA	Inhalation of isotonic pH neutral non-irritant solution prior to L-ASA inhalation	Intake of oral placebo resembling the ASA tablet
❖ **NSAID administration**	Bilateral application (micropipette) Single-dose of undiluted L-ASA several progressively increasing L-ASA doses	Inhalation using a dosimeter at tidal breathing or through counted deep breaths It is recommended the inhalation of several increasing L-ASA doses	Intake of several increasing ASA doses orally up to a therapeutic dose
❖ **Monitoring**	Combination of subjective evaluation (symptom score: visual analogue scale, total nasal symptom score, Lebel or Linder score) and objective evaluation (nasal patency: acoustic rhinometry, active anterior rhinomanometry, 4-phase Rhinomanometry or PNIF)	FEV1 as measured by forced spirometry (alternative impulse oscillometry indexes) Inflammatory biomarkers (utility in some cases)	FEV1 as measured by forced spirometry (alternative impulse oscillometry indexes) Inflammatory biomarkers (utility in some cases)
**Limitations**	- Nasal hyper-reactivity - Need for a relatively preserved nasal anatomy	- Bronchoconstriction induced by bronchial challenge - Transient increase in symptoms	- Bronchoconstriction induced by the bronchial challenge - Transient increase in symptoms
**Safety**	Extremely safe technique Late reactions are very rare After a positive test, it is recommended an observation period of 1 h at the hospital	Generally well-tolerated Late reactions can occur After a positive test, it is recommended an observation period of 7 h at the hospital	Severe and late reactions can occur. After a positive test, it is recommended an observation period of 7 h at the hospital

## Data Availability

Not applicable.

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
