# Peer review of "The Utility of Nasal Challenges to Phenotype Asthma Patients"

_ijms, 2022, doi:10.3390/ijms23094838_

Round 1

Reviewer 1 Report

The problem of asthma phenotyping with the use of nasal provocative tests was taken up in this review. The common airway theory suggests that inflammation, especially in patients with asthma, affects also the upper airways. Testing for the presence / type of nasal inflammation in patients with asthma seems to be a reasonable alternative to more invasive methods like induced sputum or BAL. In this paper, the authors discussed 2 well-known types of challenges: allergen and aspirin nasal provocation tests. It should be mentioned that the assessment of reactivity to trigger is an indirect method of assessing the phenotype. In this form, however, the work has little value to the practitioner or researcher in the field. The term phenotyping in the title of the work is also misleading. Moreover evaluation of the phenotype of particularly severe asthma based on nasal allergen challenge should be assessed in a broader context, especially in the case of overlapping certain features, for example, about half of patients with N-ERD have atopy.

Moreover, while the authors admit that they have experience in performing inhaled provocation tests with inhalatory allergens, the advantages and disadvantages of the aspirin nasal test is only briefly described. I.e. it is not mentioned , that limitation to the nasal provocation test with ASA is its lower sensitivity than the oral one, as well as poor access to aspirin in the lysine form, at least in some European countries.

Other comments:

Line 49/50 it is risky to treat bronchopulmonary aspergillosis as asthma phenotype as it may be present in other disorders (see GINA)- explanation is needed

Line 123/4- asthma control is not connected acc GINA to FEV1!

Line 136/7 symptoms occur typically after 1-2 hours, it would be safer to change to 30 minutes-2 h

Line 161/3 although there are more severe asthma patients in N-ERD patients than in other phenotypes, it cannot be generalized that the majority patients with this phenotype have severe asthma.

Figure 1 is illegible and it is not known what to say. In addition, one can get the impression that only T2-dependent asthma is present in children.

The same situation is with Figure 2, it shows that only in allergy there is granulocyte infiltration and in atopy ->T2 inflammation

Figure 3. suggests that an bronchial allergen challenge is the normal diagnostic option

Figure 4. It is not known why the diagnostic algorithm includes an alternative NSAID test and there no bronchial test (only nasal as an alternative to oral). It also suggests that a nasal provocation should be started first and then, when in doubt, oral provocation should be performed. Reference to 52 (Kowalski ML et al. Allergy 2019) should be added.

Table 1. is too extensive, it should only be shortened to significant differences.1. Extend phenotyping not only to known provocative trials but to biomarkers that can be obtained by minimally invasive nasal methods

Reviewer 2 Report

it is a nice work, the only details that I want to discuss is how do you do the diagnosys of local rhinitis? How is the nasal cellularity in this patients?

Author Response

Response to Reviewer 2 Comments

it is a nice work, the only details that I want to discuss is how do you do the diagnosis of local rhinitis? How is the nasal cellularity in this patients?

We thank the reviewer for the positive feedback. In this review we have focused on the application of nasal provocations for the diagnosis of the different asthma phenotypes and not in other applications of the nasal allergen challenge. However, as mentioned, the nasal allergen challenge with allergens is useful in the diagnosis of local allergic rhinitis. We will take the suggestion into account to delve into this aspect in future revisions.